# Processing of Degraded Speech in Brain Disorders

**DOI:** 10.3390/brainsci11030394

**Published:** 2021-03-20

**Authors:** Jessica Jiang, Elia Benhamou, Sheena Waters, Jeremy C. S. Johnson, Anna Volkmer, Rimona S. Weil, Charles R. Marshall, Jason D. Warren, Chris J. D. Hardy

**Affiliations:** 1Dementia Research Centre, Department of Neurodegenerative Disease, UCL Queen Square Institute of Neurology, London WC1N 3BG, UK; jessica.jiang.18@ucl.ac.uk (J.J.); elia.benhamou.16@ucl.ac.uk (E.B.); jeremy.johnson@ucl.ac.uk (J.C.S.J.); r.weil@ucl.ac.uk (R.S.W.); charles.marshall@qmul.ac.uk (C.R.M.); jason.warren@ucl.ac.uk (J.D.W.); 2Preventive Neurology Unit, Wolfson Institute of Preventive Medicine, Queen Mary University of London, London EC1M 6BQ, UK; s.waters@qmul.ac.uk; 3Division of Psychology and Language Sciences, University College London, London WC1H 0AP, UK; a.volkmer.15@ucl.ac.uk

**Keywords:** degraded speech processing, predictive coding, primary progressive aphasia, Alzheimer’s disease, Parkinson’s disease, perceptual learning, dementia

## Abstract

The speech we hear every day is typically “degraded” by competing sounds and the idiosyncratic vocal characteristics of individual speakers. While the comprehension of “degraded” speech is normally automatic, it depends on dynamic and adaptive processing across distributed neural networks. This presents the brain with an immense computational challenge, making degraded speech processing vulnerable to a range of brain disorders. Therefore, it is likely to be a sensitive marker of neural circuit dysfunction and an index of retained neural plasticity. Considering experimental methods for studying degraded speech and factors that affect its processing in healthy individuals, we review the evidence for altered degraded speech processing in major neurodegenerative diseases, traumatic brain injury and stroke. We develop a predictive coding framework for understanding deficits of degraded speech processing in these disorders, focussing on the “language-led dementias”—the primary progressive aphasias. We conclude by considering prospects for using degraded speech as a probe of language network pathophysiology, a diagnostic tool and a target for therapeutic intervention.

## 1. Introduction

Speech is arguably the most complex of all sensory signals and yet the healthy brain processes it with an apparent ease that belies the complexities of its neurobiological and computational underpinnings. Speech signals arrive at the ears with widely varying acoustic characteristics, reflecting such factors as speech rate, morphology, and in particular, the presence of competing sounds [1,2]. The clear speech stimuli played to participants in quiet, controlled laboratory settings are very different to the speech we typically encounter in daily life, which is usually degraded in some form. Under natural listening conditions, not only does speech often compete with other sounds, but the acoustic environment is frequently changing over time, thus speech processing is inherently dynamic. In general, the processing of degraded speech entails the extraction of an intelligible message (the “signal”) despite listening conditions that adversely affect the quality of the speech in some way (the “noise”). These conditions can be broadly conceptualised as relating to external environmental factors such as background sounds, the vocal idiosyncrasies of other speakers (such as an unfamiliar accent) [3], or feedback relating to one’s own vocal productions. Understanding speech under the non-ideal listening conditions of everyday life presents a particular challenge to the damaged brain, and might constitute a cognitive “stress test” that exposes the effects of brain pathology.

Various computational models have been proposed to explain how a speech signal is normally efficiently disambiguated from auditory “noise”, entailing the extraction of specific acoustic features, phonemes, words, syntax, and ultimately, meaning [4,5,6,7,8]. Common to these models is the notion that accurate speech decoding depends on the integration of “bottom-up” processing of incoming auditory information (e.g., speech sounds) with “top-down” prior knowledge and contextual information (e.g., stored phonemes derived from one’s native language). Degraded speech signals are parsed by generic bottom-up processes that are also engaged by other complex acoustic environments during “auditory scene analysis” [9], and the high predictability of speech signals recruits top-down processes that are relatively speech-specific: these processes normally interact dynamically and reciprocally to achieve speech recognition [10]. A computationally intensive process of this kind that depends on coherent and dynamic interactions across multiple neural processing steps is likely to be highly vulnerable to the disruptive effects of brain pathologies.

### 1.1. Predictive Coding and Degraded Speech Perception

Predictive coding theory, a current highly influential theory of perception, postulates that the brain models the causes of sensory input by iteratively comparing top-down predictions to bottom-up inputs and updating those predictions to reduce the disparity between prediction and experience (i.e., to minimise prediction error) [11,12]. The brain achieves this by modelling predictions at lower-level sensory processing stages (“priors”) via top-down connections from higher-level areas [13]: the modelling involves a weighting or gain of bottom-up inputs based on their precision (variability) and their expected precision that informs the confidence of the prediction error. In neuronal terms, this error is a mismatch between the neural representations of noisy sensory input at each processing level and the predictions constructed at the processing level above it in the hierarchy. If the prediction error is significant (above noise), this will cause the brain’s model to be modified, such that it better predicts sensory input. The computational implementation of the modification process is difficult to specify in detail a priori; the associated changes of neural activity at each processing stage are likely to evolve over time, perhaps accounting for certain apparently contradictory findings in the experimental neurophysiological literature [14].

According to this predictive coding framework, degraded speech perception depends on hierarchical reciprocal processing in which each stage passes down predictions, and prediction errors (i.e., the difference between expected and heard speech) are passed up the hierarchy [4,15]. Our ability to accurately perceive degraded speech is enhanced by inferring the probability of various possible incoming messages according to context [4,16,17].

### 1.2. Neuroanatomy of Degraded Speech Processing

From a neuroanatomical perspective, it is well established that the representation and analysis of intelligible speech occur chiefly in a processing network surrounding primary auditory cortex in Heschl’s gyrus, with processing “streams” projecting ventrally along superior temporal gyrus and sulcus (STG/STS) and dorsally to inferior frontal gyrus (IFG) in the left (dominant) cerebral hemisphere [5,18,19]. Medial temporal lobe structures in the dominant hemisphere encode and retain verbal information [19,20,21], and anterior temporal polar cortex may constitute a semantic hub [22,23,24]. The reciprocal connections between association auditory regions and prefrontal cortical areas, in particular IFG [7,25,26], are essential for the top-down disambiguation of speech signals [27,28,29].

Broadly similar regions have been consistently identified in neuroimaging studies of degraded speech processing, including superior temporal sulcus/gyrus (for accent processing: [30]; altered auditory feedback: [31]; dichotic listening: [32]; noise-vocoded speech: [33,34,35]; perceptual restoration: [36]; sinewave speech: [37]; speech-in-noise: [38]; and time-compressed speech [39]) and inferior frontal gyrus (for accent processing: [30]; noise-vocoded speech: [33,34,35]; perceptual restoration: [36]) in the dominant hemisphere. Additional temporo-parietal brain regions are also engaged under challenging listening conditions [32,40,41]. Therefore, a large fronto-temporo-parietal network consolidates information across multiple processing levels (acoustic, lexical, syntactic, semantic, articulatory) to facilitate the perception of degraded speech signals [42]. Adaptation to degraded speech may be mediated partly by subcortical striato-thalamic circuitry [27]. The macro-anatomical and functional organisation of the language network suggests how predictive coding mechanisms might operate in processing degraded speech (see Figure 1). Cortical regions involved in “early” analysis of the speech signal, such as STG/STS, communicate with “higher” regions, such as IFG, that instantiate high-level predictions about degraded sensory signals. Crucially, however, both “bottom-up” perception and “top-down” processing would occur at every stage within the hierarchy, actively updating stored templates (representations or “priors”) of the auditory environment and generating prediction errors when the auditory input fails to match the prediction [43].

Techniques such as electro-encephalography (EEG) and magneto-encephalography (MEG) have revealed dynamic, oscillatory activity that synchronises neural circuits and large-scale networks [44]. By delineating feedforward and feedback influences as well as the rapid changes that attend deviant, incongruous or ambiguous stimuli, such techniques are well suited to predictive coding applications such as the processing of degraded speech. Indeed, MEG evidence suggests that induced activity in particular frequency bands may constitute signatures of underlying neural operations during the predictive decoding of speech and other sensory signals [45,46,47]: gamma oscillations (>30 Hz) are modulated as a function of sensory “surprise” (i.e., prediction error), beta oscillations (12–30 Hz) are modulated through processing steps downstream from prediction error generation (i.e., updating of top-down predictions) and alpha oscillations (8–12 Hz) reflect the precision of predictions. Past studies conducted with MEG on degraded speech perception have shown enhanced responses in the auditory cortex (STG) when input becomes intelligible, but also reduced responses in the context of prior knowledge and perceptual learning (see Section 2.4), consistent with predictive, top-down modulation from higher-order cortical areas [48,49].

Accurate and flexible understanding of speech depends critically on the capacity of speech processing circuitry to respond efficiently, dynamically, and adaptively to diverse auditory inputs in multiple contexts and environments [50]. Degraded speech processing is therefore likely to be highly vulnerable to brain diseases that target these networks, as exemplified by the primary neurodegenerative “nexopathies” that cause dementia [51]. Major dementias strike central auditory and language processing networks relatively selectively, early and saliently (see Hardy and colleagues [52] for a review). It is therefore plausible that brain diseases should manifest as impairments of degraded speech processing and should have signature profiles of impairment according to the patterns of language network damage they produce. Indeed, reduced ability to track and understand speech under varying (non-ideal) listening conditions is a major contributor to the communication difficulties that people living with dementia experience in their daily lives and is a significant challenge for the care and management of these patients. Furthermore, the nature of the speech processing difficulty (as reflected in the symptoms patients describe) varies between different forms of dementia [52]. However, the processing of degraded speech in dementias and other brain disorders remains poorly understood and we presently lack a framework for interpreting and anticipating deficits.

### 1.3. Scope of This Review

In this review, we consider how and why the processing of degraded speech is affected in some major acquired brain disorders. Experimentally, many different types of speech degradation have been employed to study degraded speech processing: we summarise some of these in Figure 2 and provide a representative review of the literature in Table 1. We next consider important factors that affect degraded speech processing in healthy individuals to provide a context for interpreting the effects of brain disease. We then review the evidence for altered processing of degraded speech in particular acquired brain disorders (Table 2). We conclude by proposing a predictive coding framework for assessing and understanding deficits of degraded speech processing in these disorders, implications for therapy and directions for further work (Figure 3).

## 2. Factors Affecting Processing of Degraded Speech in the Healthy Brain

### 2.1. Healthy Ageing

Healthy ageing importantly influences the perception of degraded speech [52,75,77,95,96,97], and an understanding of ageing effects is essential in order to interpret the impact of brain disorders, particularly those associated with neurodegenerative disease. Ageing may be associated with functionally significant changes affecting multiple stages of auditory processing, from cochlea [98], to brainstem [99], to cortex [100]. The reduced efficiency of processing degraded speech with normal ageing is likely to reflect the interaction of peripheral and central factors [101] due, for example, to slower processing or reduced ability to regulate sensory gating [97,102,103].

These alterations in auditory pathway function tend to be amplified by age-related decline in additional cognitive functions relevant to degraded speech perception. Ageing affects domains such as episodic memory, working memory, and attention [77,101,104,105]. There is evidence to suggest that older listeners rely more heavily on “top-down” cognitive mechanisms than younger listeners, compensating for the reduced fidelity of “bottom-up” auditory signal analysis [100,106,107,108,109].

### 2.2. Cognitive Factors

The auditory system is dynamic and highly integrated with cognitive function more broadly [77,110]. Executive function is accorded central importance among the general cognitive capacities that influence the speed and accuracy of degraded speech perception, interacting with more specific skills such as phonological processing [111]. The engagement of executive processing networks—including inferior frontal gyrus, inferior parietal lobule, superior temporal gyrus and insula—during effortful listening is a unifying theme in neuroimaging studies of degraded speech processing [18]. On the other hand, the ability to process degraded speech in older adults is not entirely accounted for by general cognitive capacities [112], implying additional, auditory mechanisms are also involved.

Attention, a key cognitive factor in most sensory predictive coding models, modulates the intelligibility of degraded speech, and functional magnetic resonance imaging (fMRI) research suggests that additional frontal cortical regions are recruited when listeners attend to degraded speech signals [29]. Attention is essential for encoding precision or gain: the weighting of sensory input by its reliability [113,114]. Verbal auditory working memory—the “phonological loop”—is integral to degraded speech processing [115,116,117,118], and selective attention importantly interacts with the verbal short term store to sharpen the precision of perceptual priors held in mind over an interval (for example, during articulatory rehearsal on phonological discrimination tasks: [119]). Listeners with poorer auditory working memory capacity have more difficulty understanding speech-in-noise, even after accounting for age differences and peripheral hearing loss [77,120,121]. While working memory and attention have been studied more explicitly, it is likely that a number of cognitive factors interact in processing degraded speech, and that (in the healthy brain) the usage of these cognitive resources is dynamic and adapts flexibly to a wide variety of listening conditions [111].

### 2.3. Experiential Factors

Accumulated experience of speech signals and auditory environments over the course of the lifetime leads to the development and refinement of internal models that direct predictions about auditory input, facilitating faster neural encoding and integration [122]. Certain experiential factors, such as musical training, affect the processing of degraded speech, specifically speech-in-noise [77,123]. Musical training improves a range of basic auditory skills [124,125,126,127] and auditory working memory [128] that are important to speech encoding and verbal communications such as linguistic pitch pattern processing and temporal and frequency encoding in auditory brainstem [129,130,131,132]. This could explain findings suggesting that musicians are better at perceiving speech-in-noise (whether white-noise or babble) than non-musical listeners [76,77,133,134,135,136].

Bilingual speakers have more difficulty perceiving speech-in-noise in their non-native language than their monolingual counterparts, even when they consider themselves proficient in their non-native language [137,138,139], not necessarily in low-context situations but particularly in high-context [140]. This may be due to over-reliance on bottom-up processing with reduced integration of semantic and contextual knowledge for the second language [141,142,143], relative to more efficient top-down integration in one’s native language [139].

### 2.4. Perceptual Learning

Improved accuracy of degraded speech processing is associated with sustained exposure to the stimulus [1,54,144]: this reflects perceptual learning [145]. Perceptual learning allows listeners to learn to understand speech that has deviated from expectations [146], and typically occurs automatically and within a short period of time [49,147,148]. It is likely to reflect synaptic plasticity at different levels of perceptual analysis [149,150], and (in predictive coding terms) reflects iterative fine-tuning of the internal model with increased exposure to the stimulus, leading to error minimisation and improved accuracy of future predictions about the incoming speech signal (Figure 1; [15]).

Although perceptual learning of degraded speech is strongest and most consistent if trained and tested with the same single speaker [151,152,153], with exposure to many individuals embodying a similar particular characteristic (e.g., similar accent), the enhanced processing of that characteristic generalises to different speakers [154,155,156,157]. Longer training (i.e., more exposure to the stimulus) also leads to more stable learning and generalization [158]. Listener factors also affect perceptual learning, including language background [159], age [160], attentional set [161], and the recruitment of language processes in higher-level brain regions and connectivity [144]. Perceptual learning of accented speech in non-native listeners has been associated with improved speech production [162]. Overall, the results from studies on auditory perceptual learning suggest that it arises from dynamic interactions between different levels of the auditory processing hierarchy [163].

### 2.5. Speech Production

The functional consequences of degraded speech processing on communication cannot be fully appreciated without considering how perceptual alterations influence speech output. In the healthy brain, there is an intimate interplay between speech input and output processing, both functionally and neuroanatomically [164,165]: brain disorders that disturb this interplay are likely to have profound consequences for degraded speech processing. Speech production relies on feedback and feedforward control [166], and artificially altering auditory feedback (i.e., causing prediction errors about online feedback of one’s own speech output) frequently disrupts the speech production process [167] (see Table 1). “Altered auditory feedback” (AAF) is the collective term for auditory feedback that is altered or degraded in some manner before being played back to the speaker in real time [167], and encompasses masking auditory feedback (MAF), intensity-altered auditory feedback (IAF), delayed auditory feedback (DAF), and frequency-altered feedback (FAF). Typically, speakers will adjust their speech output automatically in some way to compensate for the altered feedback. One classical example is the “Lombard effect”, whereby the talker responds to a loud or otherwise acoustically competing environment by altering the intensity, pitch, and spectral properties of their voice [168]. Functional neuroimaging studies show that when auditory feedback is altered, there is an increase in activation in the superior temporal cortex, extending into posterior-medial auditory areas [31,169]. This corroborates other work suggesting that this region has a prominent role in sensorimotor integration and error detection [49,170].

## 3. Processing of Degraded Speech in Brain Disorders

The various factors that affect the processing of degraded speech in the healthy brain are all potentially impacted by brain diseases. Brain disorders often affect executive function, speech production, perceptual learning and other general cognitive capacities, with many becoming more frequent with age and their expression may be heavily modified by life experience.

We now consider some acquired neurological conditions that are associated with particular profiles of degraded speech processing; key studies are summarised in Table 2. While this is by no means an exhaustive list, it represents a survey of disorders that have been most widely studied and illustrates important pathophysiological principles.

### 3.1. Traumatic Brain Injury

Traumatic brain injury (TBI) refers to any alteration in brain function or structure caused by an external physical force. It therefore encompasses a wide spectrum of insults, pathological mechanisms and transient and permanent cognitive deficits [171,172]. Individuals with TBI, whether mild or severe, commonly report auditory complaints; blast-related TBI is associated with hearing loss and tinnitus in as many as 60% of patients [173]. Most data have been amassed for military veterans, and concurrent mental health issues complicate the picture [174].

People with TBI frequently report difficulties understanding speech under challenging listening conditions and a variety of central auditory deficits have been documented, including impaired speech-in-noise perception and dichotic listening [80,81,175,176]; these deficits may manifest despite normal peripheral hearing (pure tone perception), may follow mild as well as more severe injuries and may persist for years [81,82]. The culprit lesions in these cases are likely to be anatomically heterogeneous; blast exposure, for example, potentially damages auditory brainstem and cortices, corpus callosum and frontal cortex, while the preponderance of abnormal long-latency auditory evoked potentials argues for a cortical substrate [174]. Abnormal sensory gating has been proposed as an electrophysiological mechanism of impaired degraded speech processing in blast-associated TBI [83].

### 3.2. Stroke Aphasia

A number of abnormalities of degraded speech processing have been described in the context of aphasia following stroke. People with different forms of stroke-related aphasia have difficulties comprehending sentences spoken in an unfamiliar accent [85]. As might be anticipated, the profile is influenced by the type of aphasia (vascular insult) and the nature of the degraded speech manipulation: individuals with conduction aphasia and Wernicke’s aphasia show a significantly smaller benefit from DAF than people with Broca’s aphasia [177,178], while MAF was shown to improve speech rate and reduce dysfluency prolongations [86]. In patients with insular stroke, five of eight patients showed an abnormal dichotic digits test [84], and single case studies have demonstrated that people with stroke-related aphasia may have difficulty perceiving synthetic sentences with competing messages [179]. Together, these observations suggest that “informational masking” (Figure 2C) may be particularly disruptive to speech perception in stroke-related aphasia.

### 3.3. Parkinson’s Disease

Parkinson’s disease (PD), a neurodegenerative disorder caused primarily by the loss of dopaminergic neurons from the basal ganglia, is typically led by “extrapyramidal” motor symptoms including tremor, bradykinesia, and rigidity [180,181]. However, cognitive deficits are common in PD, with dementia affecting 50% of patients within 10 years of diagnosis [182]. The majority (70–90%) of individuals with PD also develop motor speech impairment [183]. Although PD is associated with objective hypophonia, people with PD overestimate the loudness of their own speech while they are speaking and in playback [184], and this is thought to be the mechanism of hypophonia due to impaired vocal feedback [185]. Responses to AAF paint a complex picture: whereas patients with PD may fail to modulate their own vocal volume under intensity altered auditory feedback [186], FAF may elicit significantly larger compensatory responses in people with PD than in healthy controls [87,88,180,187,188], while DAF substantially improves speech intelligibility in some patients with PD [189]. FAF has differential effects according to whether the fundamental frequency or the first formant of the speech signal is altered [188], and the response to altered fundamental frequency correlates with voice pitch variability [180], suggesting that the response to AAF in PD is exquisitely dependent on the nature of the perturbation and its associated sensorimotor mapping. These effects could be interpreted as specific deficits in the predictive coding of auditory information, with impaired salience monitoring as well as over-reliance on sensory priors [190,191].

Taken together, the available evidence points to abnormal auditory-motor integration in PD that tends to impair the perception of degraded speech and to promote dysfunctional communication under challenging listening conditions. Candidate neuroanatomical substrates have been identified: enhanced evoked (P2) potentials in response to FAF in PD relative to healthy controls have been localised to activity in left superior and inferior frontal gyrus, premotor cortex, inferior parietal lobule, and superior temporal gyrus [180].

### 3.4. Alzheimer’s Disease

Alzheimer’s disease (AD), the most common form of dementia, is typically considered to be an amnestic clinical syndrome underpinned by the degeneration of posterior hippocampus, entorhinal cortex, posterior cingulate, medial and lateral parietal regions within the so-called “default mode network” [192,193]. People with AD have particular difficulty with dichotic digit identification tasks [89,194,195,196]. This is likely to reflect a more fundamental impairment of auditory scene analysis that also compromises speech-in-noise and speech-in-babble perception [90,197]. During the perception of their own name over background babble (the classical “cocktail party effect”), patients with AD were shown to have abnormally enhanced activation relative to healthy older controls in right supramarginal gyrus [90]. Auditory scene analysis deficits are most striking in posterior cortical atrophy, the variant AD syndrome led by visuo-spatial impairment, further suggesting that posterior cortical regions within the core temporo-parietal network targeted by AD pathology play a critical pathophysiological role [198]. Speech-in-noise processing deficits may precede the onset of other symptoms in AD and may be a prodromal marker [199,200,201].

People with AD have difficulty understanding non-native accents [92,202] and sinewave speech (Figure 2G) [94] relative to healthy older individuals, and this has been linked using voxel-based morphometry to grey matter loss in left superior temporal cortex. Considered together with impairments of auditory scene analysis in AD, these findings could be interpreted to signify a fundamental lesion of the neural mechanisms that map degraded speech signals onto stored “templates” representing canonical auditory objects, such as phonemes. However, perceptual learning of sinewave speech has been shown to be intact in AD [94], and the comprehension of sinewave speech improves following the administration of an acetylcholinesterase inhibitor [203]. People with mild to moderate AD also show enhanced compensatory responses to FAF compared to age-matched controls [91]: this has been linked to reduced prefrontal activation and enhanced recruitment of right temporal cortices [204].

### 3.5. Primary Progressive Aphasia

Speech and language problems are leading features of the primary progressive aphasias (PPA). These “language-led dementias” constitute a heterogeneous group of disorders, comprising three cardinal clinico-anatomical syndromic variants. The nonfluent/agrammatic variant (nfvPPA) is characterised by disrupted speech and connected language production due to selective degeneration of a peri-Sylvian network centred on inferior frontal cortex and insula; the phenotype is quite variable between individual patients [205]. The semantic variant (svPPA) is characterised by the erosion of semantic memory due to selective degeneration of the semantic appraisal network in the antero-mesial (and particularly, the dominant) temporal lobe. The logopenic variant (lvPPA) is the language-led variant of AD and is characterised by anomia and impaired phonological working memory due to the degeneration of dominant temporo-parietal circuitry overlapping the circuits that are targeted in other AD variants [205,206]. All three major PPA syndromes have been shown to have clinically significant impairments of central auditory processing affecting speech comprehension [52,207,208,209,210,211,212]: together, these disorders constitute a paradigm for selective language network vulnerability and the impaired processing of degraded speech.

While people with AD have relatively greater difficulty processing less familiar non-native accents, particularly at the level of phrases and sentences, those with nfvPPA show a more pervasive pattern of impairment affecting more and less familiar accents at the level of single words [92]. People with nfvPPA and lvPPA show impaired understanding of sinewave speech relative to healthy controls and people with svPPA [94]. Patients with svPPA, however, show a significant identification advantage for more predictable (spoken number) over less predictable (spoken geographical place name) verbal signals after sinewave transformation, highlighting the important role of “top-down” contextual integration in degraded speech perception [94]. In this study, all PPA variants were shown to have intact perceptual learning of sinewave-degraded stimuli [94]. There is also evidence that at least some people with nfvPPA may be particularly susceptible to the effects of DAF [213].

The structural and functional neuroanatomy of degraded speech processing has been addressed in somewhat more detail in PPA than in other brain disorders. Using a MEG paradigm in which noise-vocoded words were presented to participants alongside written text that either matched or mismatched the degraded words, Cope and colleagues [93] found that atrophy of left inferior frontal cortex in nfvPPA was associated with inflexible and delayed neural resolution of top-down predictions about incoming degraded speech signals in the setting of enhanced fronto-temporal coherence (frontal to temporal cortical connectivity), suggesting that the process of iterative reconciliation of top-down predictions with sensory prediction error takes longer to achieve in nfvPPA. Across the nfvPPA and healthy control groups, the precision of top-down predictions correlated with the magnitude of induced beta oscillations while frontal cortical beta power was enhanced in the nfvPPA group: this is in line with predictive coding accounts according to which beta band activity reflects the updating of perceptual predictions [47]. In joint voxel-based morphometric and functional MRI studies of a combined PPA cohort [214,215], Hardy and colleagues identified a substrate for impaired decoding of spectrally degraded phonemes in left supramarginal gyrus and posterior superior temporal cortex, most strikingly in lvPPA relative to healthy older individuals, whereas nfvPPA was associated with reduced sensitivity to sound stimulation in auditory cortex. Using voxel-based morphometry in a combined AD and PPA cohort, Hardy and colleagues [94] found that the overall accuracy of sine-wave speech identification was associated with grey matter volume in left temporo-parietal cortices, with grey matter correlates of increased speech predictability in left inferior frontal gyrus, top-down semantic decoding in left temporal pole and perceptual learning in left inferolateral post-central cortex. Such studies are beginning to define the alterations in “bottom-up” and “top-down” network mechanisms that jointly underpin impaired predictive decoding of degraded speech signals in neurodegenerative disease.

## 4. A Predictive Coding Model of Degraded Speech Processing in Primary Progressive Aphasia

Emerging evidence in PPA suggests a framework for applying predictive coding theory as outlined for the healthy brain (Figure 1) to formulate explicit pathophysiological hypotheses in these diseases. Such a framework could serve as a model for interpreting abnormalities of degraded speech processing in a wider range of brain disorders. This model is outlined in Figure 3.

According to this model, nfvPPA—which affects inferior frontal and more posterior peri-Sylvian cortices—is associated with a “double-hit” to the degraded speech processing network. The most clearly established consequence is overly precise, top-down predictions due to neuronal dysfunction and loss in inferior frontal cortex [93]. The top-down mechanism may be compounded by decreased signal fidelity (precision) due to abnormal auditory cortical representations [94,214,215]; however, this remains to be corroborated. The clinico-anatomical heterogeneity of nfvPPA is an important consideration here, implying that the mechanism may not be uniform between patients.

In svPPA, the primary focus of atrophy in anterior temporal lobe principally affects the top-down integration of contextual and stored semantic information. This reduces neural capacity to modify semantic predictions about less predictable verbal signals (i.e., priors are inaccurate), in line with experimental observations [94].

In lvPPA, atrophy predominantly involving temporo-parietal cortex is anticipated to impair phonemic decoding and earlier stages in the representation of acoustic features in auditory cortex and brainstem due to altered top-down influences from the temporal parietal junction on auditory cortex and brainstem: this could be via altered precision weighting of prediction errors conveyed by the auditory efferent pathways, or inaccurate priors. This formulation has some experimental support [211,215].

## 5. Therapeutic Approaches

Improved understanding of the pathophysiology of degraded speech processing in brain disorders is the path to effective therapeutic interventions. Several physiologically informed therapeutic approaches are in current use or have shown early promise. In a clinical context, it is important not to overlook ancillary nonverbal strategies to compensate for reduced capacity to process degraded speech: examples include the minimisation of environmental noise, training speakers to face the patient to maximise visual support and aid speech sound discrimination, and using gestures to support semantic context [216,217]. A related and crucial theme in designing therapies tailored to individuals is to acknowledge the various background factors—whether deleterious or potentially protective—that influence degraded speech processing (see Section 2).

More specifically, the finding that perceptual learning of degraded speech is retained in diverse brain disorders including dementias [94] and stroke aphasia [218,219] offers the exciting prospect of designing training interventions to harness neural plasticity in these conditions. Thus far, most work in this line has been directed to improving understanding of challenging speech (in particular, speech-in-noise) in older adults with peripheral hearing loss. Training programmes have targeted different levels of speech analysis—words and sentences—and different cognitive operations—attentional and perceptuo-motor—and have shown improved perception of trained stimuli, though this is less consistently extended to untrained stimuli (the grail of work of this kind: Bieber and Gordon-Salant [220]). On the other hand, there is some evidence that training on degraded environmental sounds may generalise to improved perception of degraded speech [221]. Enhanced perceptual learning through the facilitation of regional neuronal plasticity also provides a rationale for the transcranial stimulation of key cortical language areas, such as inferior frontal gyrus [222]. Potentially, a technique such as transcranial temporal interference stimulation could selectively target deep brain circuitry and feedforward or feedback connections [223] to probe specific pathophysiological mechanisms of degraded speech processing in particular brain disorders (see Figure 3).

Other therapeutic approaches have focused on training auditory working memory. These have yielded mixed results [224], though interestingly, the training of musical working memory may show a cross over benefit for speech-in-noise recognition [225,226]. A combined auditory cognitive training programme, potentially incorporating musical skills, may be the most rational strategy [220,227].

Pharmacological approaches are potentially complementary to behavioural interventions or transcranial stimulation. In healthy individuals, dopamine has been shown to enhance the perception of spectrally shifted noise-vocoded speech [228]. In patients with AD, acetylcholinesterase inhibition ameliorates the understanding of sinewave speech [203]. Indeed, degraded speech processing might prove to be a rapid and sensitive biomarker of therapeutic efficacy in brain disorders. At present, the objectives of therapy differ quite sharply between disorders such as stroke, where there is a prospect of sustained improvement in functional adaptation in at least some patients, and neurodegenerative conditions such as PPA, where any benefit is ultimately temporary due to the progressive nature of the underlying pathology. However, it is crucial to develop interventions that enhance degraded speech processing (and other ecologically relevant aspects of communication) in neurodegenerative disease, not only to maximise patients’ daily life functioning but also with a future view to using such techniques adjunctively with disease modifying therapies as these become available. Ultimately, irrespective of the brain pathology, it will be essential to determine how far improvements on degraded speech processing tasks translate to improved communication in daily life.

## 6. A Critique of the Predictive Coding Paradigm of Degraded Speech Processing

Like any scientific paradigm, predictive coding demands a critical evaluation of falsifiable hypotheses. The issues in relation to the auditory system have been usefully reviewed previously in Heilbron and Chait [14]. While it is self-evident that the brain is engaged in making and evaluating predictions, there are two broad questions here, in respect of degraded speech processing that could address and direct future experiments.

Firstly, to what extent is the processing of degraded speech generically underpinned by predictive coding? While the predictive coding paradigm is committed to finding optimal computational solutions to perceptual perturbations, much natural language use relies on acoustic or articulatory characteristics that are “sub-optimal” [229]. More fundamentally, as the raw material is contributed much to human thought, the combinatorial space of language is essentially infinite: we routinely produce entirely novel utterances and are called upon to understand the novel utterances of others, whereas predictive coding rests on a relatively simple computational “logic” [230]. Identifying the limits of predictive coding in the face of emergent linguistic combinatorial complexity therefore presents a major challenge—a challenge encountered even for the combinatorially much more constrained phenomenon of music [231]. Future experiments will need to define core predictive coding concepts such as “priors”, “error” and “precision” in terms of degraded speech processing, as well as disambiguate the roles of semantic and phonological representations, selective attention and verbal working memory in such processing, ideally by manipulating these components independently [14,191,232,233,234].

Secondly, how is the predictive coding of degraded speech instantiated in the brain? Although macroscopic neural network substrates that could support the required hierarchical and reciprocal information exchange have been delineated (Figure 1), the predictive coding paradigm stipulates quite specifically how key elements such as “prediction generators” and “error detectors” are organised, both at the level of large-scale networks and local cortical circuits [14,235]. Neuroimaging techniques such as spectral dynamic causal modelling, MEG and high-field fMRI constitute particularly powerful and informative tools with which to interrogate the responsible neural elements and their interplay [14,236]: such techniques can capture both interactions between macroscopic brain modules and structure–function relationships at the level of individual cortical laminae, where the core circuit components of predictive coding are hypothesised to reside.

## 7. Conclusions and Future Directions

The perception and ultimately understanding of degraded speech relies upon flexible and dynamic neural interactions across distributed brain networks. These physiological and anatomical substrates are intrinsically vulnerable to the disruptive effects of brain disorders, particularly neurodegenerative pathologies that preferentially blight the core circuitry responsible for representing and decoding speech signals. Predictive coding offers an intuitive framework within which to consider degraded speech processing, both in the healthy brain (Figure 1) and in brain disorders (Figure 3). Different forms of speech signal degradation are likely a priori to engage neural network nodes and connections differentially and may therefore reveal distinct phenotypes of degraded speech processing that are specific for particular neuropathological processes. However, this will require substantiation in future systematic, head-to-head comparisons between paradigms (Table 1, Figure 2) and pathologies (Table 2, Figure 3). It will be particularly pertinent to design neuropsychological and neuroimaging experiments to interrogate the basic assumptions of predictive coding theory, as sketched above.

From a neurobiological perspective, building on the model outlined for PPA in Figure 3, degraded speech is an attractive candidate probe of pathophysiological mechanisms in brain disease. For example, it has been proposed that lvPPA is associated with the “blurring” of phonemic representational boundaries [211]: this would predict that phonemic restoration (Figure 2) is critically impaired in lvPPA. Further, several lines of evidence implicate disordered efferent regulation of auditory signal analysis in the pathogenesis of nfvPPA [93,210]: this could be explored directly by independently varying the precision of incoming speech signals and central gain (for example, using dichotic listening techniques). Temporally sensitive neurophysiological and functional neuroimaging techniques such as EEG and MEG will be required to define the dynamic oscillatory neural mechanisms by which brain pathologies disrupt degraded speech perception. Proteinopathies are anticipated to have separable MEG signatures based on differential patterns of cortical laminar involvement [237]. By extension from the “lesion studies” of classical neurolinguistics, the study of clinical disorders may ultimately illuminate the cognitive and neural organisation of degraded speech processing in the normal brain [93], by pinpointing critical elements and demonstrating how dissociable processing steps are mutually related.

From a clinical perspective, the processing of degraded speech (as a sensitive index of neural circuit integrity) might facilitate the early diagnosis of brain disorders. Neurodegenerative pathologies, in particular, often elude diagnosis in their early stages: degraded speech stimuli might be adapted to constitute dynamic, physiological “stress tests” to detect such pathologies. Similar pathophysiological principles should inform the design of behavioural and pharmacological therapies, such as those that harness neural plasticity: looking forward, such interventions could be particularly powerful if combined with disease modifying therapies, as integrated cognitive neurorehabilitation strategies motivated by neurobiological principles.

## Figures and Tables

**Figure 1 brainsci-11-00394-f001:**
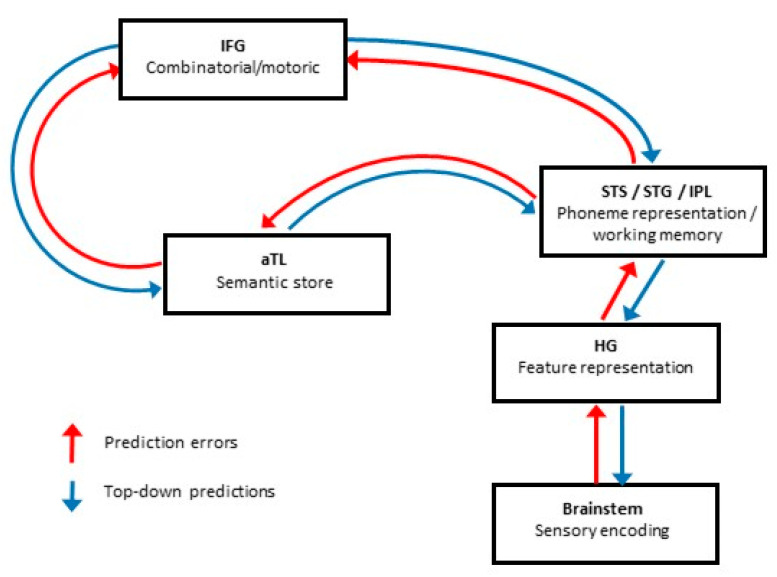
A predictive coding model of normal degraded speech processing with major anatomical loci for core speech decoding operations and their connections, informed by evidence in the healthy brain. Different kinds of degraded speech manipulation are likely to engage these cognitive operations and connections differentially (see Table 1). Incoming sensory information undergoes “bottom-up” perceptual analysis chiefly in early auditory areas, while higher level brain regions generate predictions about the content of the speech signal. Boxes indicate processors that instantiate core functions; note, however, that processing “levels” are not strictly confined to higher-order predictions or early sensory input: interactions occur at each level. Arrows indicate connections between levels, with reciprocal information flow mediating modulatory influences and dynamic updating/perceptual learning of degraded speech signals. This figure is necessarily an over-simplification; cortical areas that are likely to have separable functional roles are grouped together for clarity of representation, and while they are not shown in this figure, intra-areal recurrences and inhibitions alongside other local circuit effects may also be operating within these regions. aTL, anterior temporal lobe; HG, Heschl’s gyrus; IFG, inferior frontal gyrus; IPL, inferior parietal lobule; STG, superior temporal gyrus; STS, superior temporal sulcus.

**Figure 2 brainsci-11-00394-f002:**
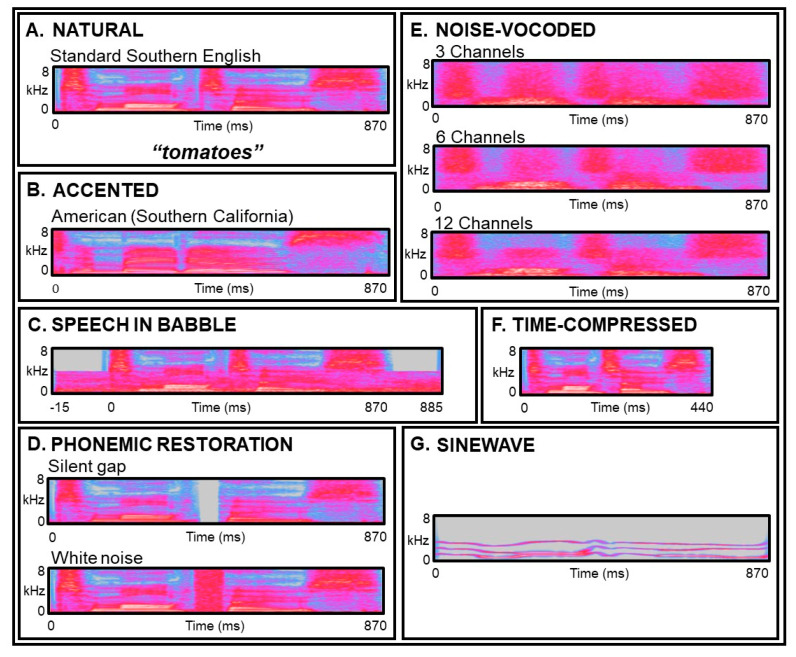
Examples of degraded speech manipulations used experimentally and their acoustic effects on the speech signal. Broadband time-frequency spectrograms of the same speech token (“tomatoes”), subjected to different forms of speech degradation (all samples apart from 2B were recorded by a native British speaker with a Standard Southern English accent; wavefiles of A–G are in Appendix A online). (**A**) Natural speech token. (**B**) Same speech token spoken with an American-Californian accent (an accent is a meta-linguistic feature that reveals information about the speaker’s geographical or socio-cultural background [53]; normal listeners make predictions about speakers’ accents that tend to facilitate faster accent processing [54]). (**C**) Speech in multi-talker babble (speech-in-noise can be adaptively adjusted to find the point at which speech switches from intelligible to unintelligible [55]; background “noise” used experimentally typically comprises either “energetic” masking (e.g., steady-state white noise) or “informational” masking (e.g., multi-talker babble, as illustrated here)) [56], (**D**) Perceptual (or phonemic) restoration (Warren [57] originally observed that when a key phoneme is artificially excised from a given sentence, control participants are unable to identify the location of the missing phoneme when “filled-in” with a burst of white noise (bottom panel), but are able to identify the location accurately if the gap remains silent (top panel), i.e., they perceptually “restore” the excised phoneme). (**E**) Noise-vocoded speech (vocoding removes fine spectral detail from speech, whilst preserving temporal cues [58,59]; three bands of modulated noise (i.e., three “channels”; top panel) are the minimum needed for consistent recognition by normal listeners [59], spectrograms for six (middle panel) and twelve (bottom panel) channels also shown here). (**F**) Time-compressed speech (created by artificially increasing the rate at which a recorded speech stimulus is presented; intelligibility decreases as speech compression rate increases [60,61,62]). (**G**) Sinewave speech (this transformation reduces speech to a series of “whistles” or sinewave tones that track formant contours [63]). Note that these speech manipulations vary widely in the cognitive process they target, the degree to which they degrade the speech signal and their ecological resonance (see also Table 1); accented speech and speech-in-noise or babble are commonly encountered in daily life through exposure to diverse speakers and noisy environments, perceptual restoration simulates the frequent everyday phenomenon of speech interruption by intermittent extraneous sounds (e.g., a slamming door), whereas sinewave-speech is a drastic impoverishment of the speech signal that sounds highly unnatural but becomes intelligible with exposure due to perceptual learning [64].

**Figure 3 brainsci-11-00394-f003:**
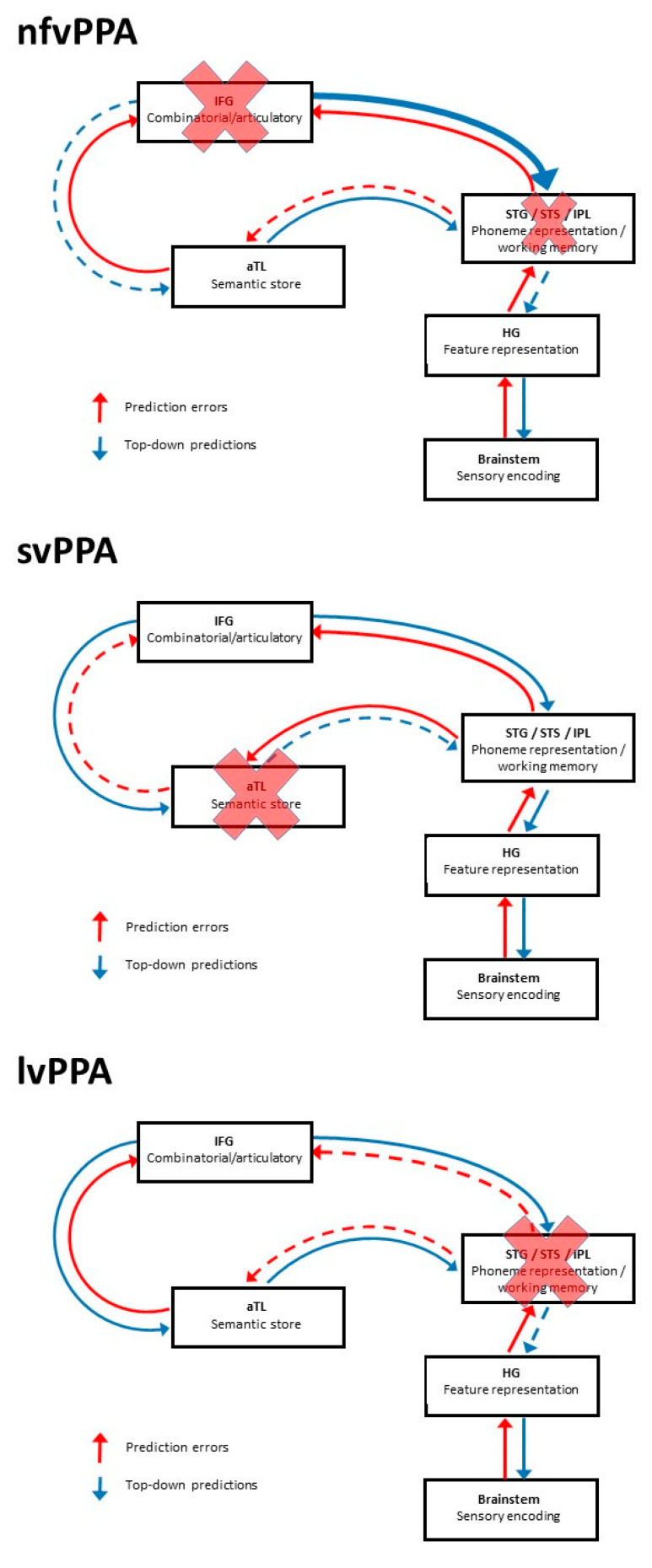
A simplified model of predictive coding of degraded speech processing in primary progressive aphasia (PPA), referenced to the healthy brain presented in Figure 1. The three major PPA variant syndromes—nonfluent/agrammatic variant PPA (top panel); semantic variant PPA (middle panel) and logopenic variant PPA (bottom panel)—are each associated with a specific pattern of regional brain atrophy and/or dysfunction that is critical to the degraded speech processing network, implying that different PPA subtypes may be associated with specific profiles of degraded speech processing (see text for details). Boxes indicate processors that instantiate core speech decoding functions (see Figure 1), and arrows indicate their connections in the predictive coding framework, with the putative direction of information flow. In the case of nfvPPA, the emboldened descending arrow from IFG to STG signifies aberrantly increased precision of inflexible top-down priors (after Cope and Colleagues [93]), to date the most secure evidence for a predictive coding mechanism in the PPA spectrum; the status of the IPL locus in this syndrome is more tentative. Implicit in the model is the hypothesis that neurodegenerative pathologies will tend to disrupt stored neural templates (“priors”) and “prune” projections from heavily involved, higher order association cortical areas due to neuronal dropout (promoting inflexible top-down predictions), but also degrade the fidelity of signal traffic through sensory cortices (reducing sensory precision and promoting over-precise prediction errors) [15]. The relative prominence of these mechanisms will depend on the macro-network and local neural circuit anatomy of particular neurodegenerative pathologies. Proposed major loci of disruption caused by each PPA variant are indicated with crosses; dashed arrows arising from these damaged modules indicate disrupted information flow. aTL, anterior temporal lobe; HG, Heschl’s gyrus; IFG, inferior frontal gyrus; IPL, inferior parietal lobule; lvPPA, logopenic variant primary progressive aphasia; nfvPPA, non-fluent variant primary progressive aphasia; STG superior temporal gyrus; STS, superior temporal sulcus; svPPA, semantic variant primary progressive aphasia.

**Table 1 brainsci-11-00394-t001:** Summary of major forms of speech degradation with representative experimental studies in healthy listeners.

Degradation Type	Study	Participants	Methodology	Major Findings
**ACCENTS** **Target process:** phonemic and intonational representations **Ecological relevance:** Understanding messages conveyed via non-canonical spoken phonemes and suprasegmental intonation	Bent and Bradlow [65]	65 healthy participants (age: 19.1)	Participants listened to English sentences spoken by Chinese, Korean, and English native speakers.	Non-native listeners found speech from non-native English speakers as intelligible as from a native speaker.
	Clarke and Garrett [66]	164 healthy participants (American English)	Participants listened to English sentences spoken with a Spanish, Chinese, and English accent.	Processing speed initially slower for accented speech, but this deficit diminished with exposure.
	Floccia, Butler, Goslin and Ellis [54]	54 healthy participants (age 19.7; Southern British English)	Participants had to say if the last word in a spoken sentence was real or not.	Changing accent caused a delay in word identification, whether accent change was regional or foreign.
**ALTERED AUDITORY FEEDBACK** **Target process:** Influence of auditory feedback on speech production **Ecological relevance:** Ability to hear, process, and regulate speech from own production.	Siegel and Pick [67]	20 healthy participants	Participants produced speech whilst hearing amplified feedback of their own voice.	Participants lowered their voices (displaying the sidetone amplification effect) in all conditions.
	Jones and Munhall [68]	18 healthy participants (age: 22.4; Canadian English)	Participants produced vowels with altered feedback of F0 shifted up or down.	Participants compensated for change in F0.
	Donath et al. [69]	22 healthy participants (age: 23; German)	Participants said a nonsense word with feedback of their frequency randomly shifting downwards.	Participants adjusted their voice F0 after a set period of time due to processing the feedback first.
	Stuart et al. [70]	17 healthy participants (age: 32.9; American English)	Participants spoke under DAF at 0, 25, 50, 200 ms at normal and fast rates of speech.	There were more dysfluencies at 200 ms, and more dysfluencies at the fast rate of speech.
**DICHOTIC LISTENING** **Target process:** Auditory scene analysis (auditory attention) **Ecological relevance:** Processing of spoken information with competing verbal material	Moray [71]	Healthy participants, no other information given	Participants were told to focus on a message played to one ear, with a competing message in the other ear.	Participants did not recognize the content in the unattended message.
	Lewis [72]	12 healthy participants	Participants were told to attend to message presented in one ear, with a competing message in the other.	Participants could not recall the unattended message, but semantic similarity affected reaction times.
	Ding and Simon [73]	10 healthy participants (age 19–25)	Under MEG, participants heard competing messages in each ear, and asked to attend to each in turn.	Auditory cortex tracked temporal modulations of both signals, but was stronger for the attended one.
**NOISE-VOCODED SPEECH** **Target process:** Phonemic spectral detail **Ecological relevance:** Understanding whisper (similar quality to speech heard by cochlear implant users)	Shannon, Zeng, Kamath, Wygonski and Ekelid [59]	8 healthy participants	Participants listened to and repeated simple sentences that had been noise-vocoded to different degrees.	Performance improved with number of channels; high speech recognition was achieved with only 3 channels.
	Davis, Johnsrude, Hervais-Adelman, Taylor and McGettigan [58]	12 healthy participants (age 18–25; British English)	Participants listened to and then transcribed 6-channel noise-vocoded sentences.	Participants showed rapid improvement over the course of 30-sentence exposure.
	Scott, Rosen, Lang and Wise [35]	7 healthy participants (age 38)	Under PET, participants listened to spoken sentences that were noise-vocoded to various degrees.	Selective response to speech intelligibility in left anterior STS.
**PERCEPTUAL RESTORATION** **Target process:** Message interpolation **Ecological relevance:** Understanding messages in intermittent or varying noise (e.g., a poor telephone line)	Warren [57]	20 healthy participants	Participants identified where the gap was in sentences where a phoneme was replaced by silence/white noise.	Participants were more likely to mislocalize a missing phoneme that was replaced by noise.
	Samuel [74]	20 healthy participants (English)	Participants heard sentences in which white noise was either “Added” to or “Replaced” a phoneme.	Phonemic restoration was more common for longer words and certain phone classes.
	Leonard, Baud, Sjerps and Chang [43]	5 healthy participants (age 38.6; English/Italian)	Subdural electrode arrays recorded while participants listened to words with noise-replaced phonemes.	Electrode responses were comparable to intact words vs. words with a phoneme replaced.
**SINEWAVE SPEECH** **Target process:** Speech reconstruction and adaptation from very impoverished cues **Ecological relevance:** Synthetic model for impoverished speech signal and perceptual learning	Remez, Rubin, Pisoni and Carrell [63]	54 control participants	Naïve listeners heard SWS replicas of spoken sentences and were later asked to transcribe the sentences.	Most listeners did not initially identify the SWS as speech, but were able to transcribe them when told this.
	Barker and Cooke [64]	12 control participants	Participants were asked to transcribe SWS or amplitude-comodulated SWS sentences.	Recognition for SWS ranged from 35–90%, and amplitude-comodulated SWS ranged from 50–95%.
	Möttönen, Calvert, Jääskeläinen, Matthews, Thesen, Tuomainen and Sams [37]	21 control participants (18–36; English)	Participants underwent two fMRI scans: one before training on SWS, and one post-training.	Activity in left posterior STS was increased after SWS training.
**SPEECH-IN-NOISE** **Target process:** Auditory scene analysis (parsing of phonemes from acoustic background) **Ecological relevance:** Understanding messages in background noise (e.g., “cocktail party effect”)	Pichora-Fuller et al. [75]	24 participants in three groups (age 23.9; 70.4; 75.8; English)	Participants repeated the last word of sentences in 8-talker babble. Half had predictable endings.	Both groups of older listeners derived more benefit from context than younger listeners.
	Parbery-Clark et al. [76]	31 control participants (incl. 16 musicians; age: 23; English)	Participants were assessed via clinical measures of speech perception in noise.	Musicians outperformed the non-musicians on both QuickSIN and HINT.
	Anderson et al. [77]	120 control participants (age 63.9)	Peripheral auditory function, cognitive ability, speech-in-noise, and life experience were examined.	Central processing and cognitive function predicted variance in speech-in-noise perception.
**TIME-COMPRESSED SPEECH** **Target process:** Phoneme duration (rate of presentation) **Ecological relevance:** Understanding rapid speech	Dupoux and Green [60]	160 control participants (English)	Participants transcribed spoken sentence were compressed to 38% and 45% of their original durations.	Participants improved over time. This happened more rapidly for the 45% compressed sentences.
	Poldrack et al. [78]	8 control participants (age: 20–29; English)	Participants listened to time-compressed speech. Brain responses were tracked using fMRI.	Activity in bilateral IFG and left STG increased with compression, until speech became incomprehensible.
	Peelle et al. [79]	8 control participants (age: 22.6; English)	Participants listened to sentences manipulated for complexity and time-compression in an fMRI study.	Time-compressed sentences recruited AC and premotor cortex, regardless of complexity.

The table is ordered by type of speech degradation. Information in the Participants column is based on available information from the original papers; age is given as a mean or range and language refers to participants’ native languages. Abbreviations: AC, anterior cingulate; DAF, delayed auditory feedback; F0, fundamental frequency; fMRI; functional magnetic resonance imaging; HINT, Hearing in Noise Test; IFG, inferior frontal gyrus; ms, millisecond; QuickSIN, Quick Speech in Noise Test; PET, positron emission tomography; STG, superior temporal gyrus; STS, superior temporal sulcus; SWS, sinewave speech.

**Table 2 brainsci-11-00394-t002:** Summary of representative studies of degraded speech processing in clinical populations.

Population	Study, Degradation	Participants	Methodology	Major Findings
Traumatic brain injury	Gallun et al. [80]: Central auditory processing	36 blast-exposed military veterans (age: 32.8); 29 controls (age: 32.1)	Participants went through a battery of standardised behavioural tests of central auditory function: temporal pattern perception, GIN, MLD, DDT, SSW, and QuickSIN.	While no participant performed poorly on all behavioural testing, performance was impaired in central auditory processing for the blast-exposed veterans in comparison to matched-controls.
	Saunders et al. [81]: Central auditory processing	99 military veterans (age: 34.1)	Participants went through self-reported measures as well as a battery of standardised behavioural measures: HINT, NA LiSN-S, ATTR, TCST, and SSW.	Participants in this study showed measurable performance deficits on speech-in-noise perception, binaural processing, temporal resolution, and speech segregation.
	Gallun et al. [82]: Central auditory processing	30 blast-exposed military veterans, with a least one blast occurring 10 years prior to study (age: 37.3); 29 controls (age: 39.2)	Participants went through a battery of standardised behavioural tests of central auditory function: GIN, DDT, SSW, FPT, and MLD.	Replicating the findings from Gallun et al., 2012, this study found that the central auditory processing deficits persisted in individuals tested an average of more than 7 years after blast exposure.
	Papesh et al. [83]: Central auditory processing	16 blast-exposed veterans (age 36.9); 13 veteran controls (age 38) with normal peripheral hearing	Participants competed self-reported measures and standardised tests of speech-in-noise perception, DDT, SSW, TCST, plus auditory event-related potential studies.	Impaired cortical sensory gating was primarily influenced by a diagnosis of TBI and reduced habituation by a diagnosis of post-traumatic stress disorder. Cortical sensory gating and habituation to acoustic startle strongly predicted degraded speech perception
Stroke aphasia	Bamiou et al. [84]: Dichotic listening	8 patients with insular strokes (age: 63); 8 control participants (age: 63)	Participants heard pairs of spoken digits presented simultaneously to each ear, and were asked to repeat all four digits.	Dichotic listening was abnormal in five of the eight stroke patients.
	Dunton et al. [85]: Accents	16 participants with aphasia (age: 59); 16 controls (age: 59; English)	Participants heard English sentences spoken with a familiar (South-East British England) or unfamiliar (Nigerian) accent.	Aphasia patients made more errors in comprehending sentences spoken in an unfamiliar accent vs. a familiar accent.
	Jacks and Haley [86]: AAF (MAF)	10 aphasia patients (age: 53.1); 10 controls (age: 63.1; English)	Participants produced spoken sentences with no feedback, DAF, FAF or noise-masked auditory feedback (MAF).	Speech rate increased under MAF but decreased with DAF and FAF in most participants with aphasia.
Parkinson’s disease	Liu et al. [87]: AAF (MAF and FAF)	12 PD participants (ge: 62.3); 13 control participants (age: 68.7)	Participants sustained a vowel whilst receiving changes in feedback of loudness (±3/4 dB) or pitch (±100 cents).	All participants produced compensatory responses to AAF, but response sizes were larger in PD than controls.
	Chen et al. [88]: AAF (FAF)	15 people with PD (age: 61); 15 control participants (age 61; Cantonese)	Participants were asked to vocalize a vowel sound with AAF pitch-shifted upwards or downwards.	PD participants produced larger magnitudes of compensation.
Alzheimer’s disease	Gates et al. [89]: Dichotic digits	17 ADs (age: 84); 64 MCI (age: 82.3); 232 controls (age: 78.8)	Participants listened to 40 numbers presented in pairs to each ear simultaneously.	AD patients scored the worst in the dichotic digits, followed by the MCI group and then the controls.
	Golden et al. [90]: Auditory scene analysis	13 AD participants (age: 66); 17 control participants (age: 68)	In fMRI, participants listened to their own name interleaved with or superimposed on multi-talker babble.	Significantly enhanced activation of right supramarginal gyrus in the AD vs. control group for the cocktail party effect.
	Ranasinghe et al. [91]: AAF (FAF)	19 AD participants; 16 control participants	Participants were asked to produce a spoken vowel in context of AAF, with perturbations of pitch.	AD patients showed enhanced compensatory response and poorer pitch-response persistence vs. controls.
Primary progressive aphasia	Hailstone et al. [92]: Accents	20 ADs (age: 66.4); 6 nfvPPA (age: 66); 35 controls (age: 65); British English	Accent comprehension and accent recognition was assessed. VBM examined grey matter correlates.	Reduced comprehension for phrases in unfamiliar vs. familiar accents in AD and for words in nfvPPA; in AD group, grey matter associations of accent comprehension and recognition in anterior superior temporal lobe
	Cope et al. [93]: Noise-vocoding	11 nfvPPA (age: 72); 11 control participants (age: 72)	During MEG, participants listened to vocoded words presented with written text that matched/mismatched.	People with nfvPPA compared to controls showed delayed resolution of predictions in temporal lobe, enhanced frontal beta power and top-down fronto-temporal connectivity; precision of predictions correlated with beta power across groups
	Hardy et al. [94]: SWS	9 nfvPPA (age: 69.6); 10 svPPA (age: 64.9); 7 lvPPA (age: 66.3); 17 control (age: 67.7)	Participants transcribed SWS of numbers/locations. VBM examined grey matter correlates in combined patient cohort.	Variable task performance groups; all showed spontaneous perceptual learning effects for SWS numbers; grey matter correlates in a distributed left hemisphere network extending beyond classical speech-processing cortices, perceptual learning effect in left inferior parietal cortex

Information in the Participants column is based on available information from the original papers; age is given as a mean or range and language refers to participants’ native languages. Abbreviations: AAF, altered auditory feedback; AD, Alzheimer’s disease; ATTR, Adaptive Tests of Temporal Resolution; DAF, delayed auditory feedback; dB, decibels; DDT, Dichotic Digits Test; FAF; frequency altered feedback; fMRI, functional magnetic resonance imaging; FPT, Frequency Patterns Tests (FPT); GIN, Gaps-In-Noise test; HINT, Hearing in Noise Test; lvPPA, logopenic variant primary progressive aphasia; MAF, masked/masking auditory feedback; MCI, mild cognitive impairment; MEG, magnetoencephalography; MLD, The Masking Level Difference; NA LiSN-S, North American Listening in Spatialised Noise-Sentence test; nfvPPA, nonfluent primary progressive aphasia; PD, Parkinson’s disease; PR, perceptual restoration; QuickSIN, Quick Speech in Noise; SSW, Staggered Spondaic Words; SWS, sinewave speech; svPPA, semantic variant primary progressive aphasia; TBI, traumatic brain injury; TCST, Time Compressed Speech Test; VBM, voxel based morphometry.

## Data Availability

Not applicable.

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
