# Peer review of "Processing of Degraded Speech in Brain Disorders"

_brainsci, 2021, doi:10.3390/brainsci11030394_

Round 1
Reviewer 1 Report
The current manuscript does a great job of presenting the possibility of using degraded speech comprehension abilities to differentiate between different neurological disorders. I appreciated the detailed review including the tables that provided a good background for the study objectives. It was also helpful to see different sections discussing each of the neurological disorders as each disorder is often characterized by specific brain changes and effects. Overall, the manuscript does a great job of explaining how degraded speech can be an effective tool to better distinguish different clinical forms of neurological disorders such as primary progressive aphasia.
Author Response
We thank Reviewer #1 for their comments and support for this manuscript
Reviewer 2 Report
Dear Authors,
i read your manuscript and i have no comments or suggestions to do for your work, since it is a very good contribution.
Author Response
We thank Reviewer #2 for their comments and support for this manuscript.
Reviewer 3 Report
This paper reviews the literature on an important topic, namely the ways in which degraded speech is perceived and processed in several relevant brain disorders. Additionally, it proposes a predictive coding framework, focusing largely on the various subtypes of primary progressive aphasia. The manuscript reads very well and presents a nice summary of some key issues and representative papers for the aspects focused on. In terms of suggestions for improvement, there are just a few ways in which things could be refined, or more appropriately supported.
The only notable concern that I would raise is the label of the anatomical site in Figure 1 for phoneme representation. The authors note several anatomical models for speech processing (Davis and Sohoglu, 2019; Hickok and Poeppel, 2007; Okada et al., 2010; Peelle, 2010; Scott et al., 2000) but almost all of these models posit the STS as the primary locus of phonemic representation, with relatively less focus on STG, and none of the papers focusing on the TPJ. So why the STG/TPJ label? It is recommended that this be changed to the more agreed upon STS, but if STG/TPJ is key to the authors arguments, then this anatomical locus needs to be strongly supported with relevant citations demonstrating this point.
A few minor points are that there did not seem to be a sufficient justification for the particular patient populations used. It would seem that others (e.g. TBI) could also be included, and an explanation for what makes the included groups particularly vulnerable to impairments in degraded speech processing would be helpful. Similarly, though most of the paper seems to focus on disorders as they affect speech perception, there are short sections on normal aging, and on speech production, and it might be worth expanding these, or at least making their relevance to the main text a bit more explicit. The other minor point was that on pg. 3, the use of double parentheses rather than brackets or another form of punctuation detracts a bit from the clarity of that section.
Author Response
THE ONLY NOTABLE CONCERN THAT I WOULD RAISE IS THE LABEL OF THE ANATOMICAL SITE IN FIGURE 1 FOR PHONEME REPRESENTATION. THE AUTHORS NOTE SEVERAL ANATOMICAL MODELS FOR SPEECH PROCESSING (DAVIS AND SOHOGLU, 2019; HICKOK AND POEPPEL, 2007; OKADA ET AL., 2010; PEELLE, 2010; SCOTT ET AL., 2000) BUT ALMOST ALL OF THESE MODELS POSITS THE STS AS THE PRIMARY LOCUS OF PHONEMIC REPRESENTATION, WITH RELATIVELY LESS FOCUS ON STG, AND NONE OF THE PAPERS FOCUSSING ON THE TPJ. SO WHY THE STG/TPJ LABEL? IT IS RECOMMENDED THAT THIS BE CHANGED TO THE MORE AGREED UPON STS, BUT IF STG/TPJ IS KEY TO THE AUTHOURS ARGUMENTS, THEN THIS ANATOMICAL LOCUS NEEDS TO BE STRONGLY SUPPORTED WITH RELEVANT CITATIONS DEMONSTRATING THIS POINT.
We agree with the Reviewer concerning the involvement of STS in phoneme representation. In the revised Figures (Figure 1, pg. 3; Figure 3, pg. 12) we now include STS, alongside STG, for phoneme representation. We have also introduced ‘IPL’, specifically to cover the likely role of phonemic working memory in degraded speech processing and removed the rather vague designation ‘TPJ’.
Specific changes:
- Image change for Figure 1 on pg. 3
- Changes to the description of Figure 1 are given and highlighted (pg. 3 and 4)
- Image change for Figure 3 on pg. 12
- Changes to the description of Figure 3 are given and highlighted (pg. 13)
A FEW MINOR POINTS ARE THAT THERE DID NOT SEEM TO BE A SUFFICIENT JUSTIFICATION FOR THE PARTICULAR PATIENT POPULATIONS USED. IT WOULD SEEM THAT OTHERS (E.G., TBI) COULD ALSO BE INCLUDED, AND AN EXPLANATION FOR WHAT MAKES THE INCLUDED GROUPS PARTICULARLY VULNERABLE TO IMPAIRMENTS IN DEGRADED SPEECH PROCESSING WOULD BE HELPFUL.
We thank the Reviewer for suggesting we include traumatic brain injury (TBI) which we now do. We believe our coverage in the review reflects the rather variable emphases of the field at large and we now acknowledge this as a limitation (pg 15) and foreground more systematic study as an important direction for future work (pg 20).
Specific changes:
- Wording changes to the introduction of section 3 (pg. 15, paragraph 2)
- New added section: 3.1 Traumatic brain injury (pg. 15, paragraph 3)
- New section titles following this addition
- Added section in Table 2 for TBI studies (pg. 9)
SIMILARLY, THOUGH MOST OF THE PAPER SEEMS TO FOCUS ON DISORDERS AS THEY AFFECT SPEECH PERCEPTION, THERE ARE SHORT SECTIONS ON NORMAL AGING, AND ON SPEECH PRODUCTION, AND IT MIGHT BE WORTH EXPANDING THESE, OR AT LEAST MAKING THEIR RELEVANCE TO THE MAIN TEXT A BIT MORE EXPLICIT.
We feel that including some background information about normal aging and speech production helps the non-specialist reader by providing context for the more specific analysis of the clinical degraded speech literature. We have underlined the relevance of this background in the healthy ageing (pg. 13) and speech production (pg 15) sections.
Specific changes:
- Wording changes to section 2.1 healthy ageing (pg. 13, paragraph 1)
- Wording changes to section 2.5 speech production (pg. 15, paragraph 1)
THE OTHER MINOR POINT WAS THAT ON PG. 3, THE USE OF DOUBLE PARENTHESES RATHER THAN BRACKETS OR ANOTHER FORM OF PUNCTUATION DETRACTS A BIT FROM THE CLARITY OF THAT SECTION.
We agree and have now made the changes on (pg. 2-3).
Specific changes:
- Citation changes (pg. 2-3)